# Epigenetic Signaling and RNA Regulation in Cardiovascular Diseases

**DOI:** 10.3390/ijms21020509

**Published:** 2020-01-13

**Authors:** Alessia Mongelli, Sandra Atlante, Tiziana Bachetti, Fabio Martelli, Antonella Farsetti, Carlo Gaetano

**Affiliations:** 1Laboratorio di Epigenetica, Istituti Clinici Scientifici Maugeri IRCCS, Via Maugeri 4, 27100 Pavia, Italy; alessia.mongelli@icsmaugeri.it (A.M.); sandra.atlante@icsmaugeri.it (S.A.); 2Direzione Scientifica, Istituti Clinici Scientifici Maugeri IRCCS, Via Maugeri 4, 27100 Pavia, Italy; tiziana.bachetti@icsmaugeri.it; 3Molecular Cardiology Laboratory, Policlinico San Donato IRCCS, San Donato Milanese, 20097 Milan, Italy; fabio.martelli@grupposandonato.it; 4Institute for Systems Analysis and Computer Science “A. Ruberti” (IASI), National Research Council (CNR), 00185 Rome, Italy

**Keywords:** epigenetics, nucleic acids, RNA, DNA, cardiovascular disease, chronic disease, aging, metabolism

## Abstract

RNA epigenetics is perhaps the most recent field of interest for translational epigeneticists. RNA modifications create such an extensive network of epigenetically driven combinations whose role in physiology and pathophysiology is still far from being elucidated. Not surprisingly, some of the players determining changes in RNA structure are in common with those involved in DNA and chromatin structure regulation, while other molecules seem very specific to RNA. It is envisaged, then, that new small molecules, acting selectively on RNA epigenetic changes, will be reported soon, opening new therapeutic interventions based on the correction of the RNA epigenetic landscape. In this review, we shall summarize some aspects of RNA epigenetics limited to those in which the potential clinical translatability to cardiovascular disease is emerging.

## 1. Introduction

Translational epigenetics is a relatively new branch of molecular biology that investigates regulatory processes occurring molecularly “above” the primary DNA sequence associated with physiological and pathophysiological conditions. Specifically, of applied translational epigenetics interest are those mechanisms that introduce functional changes into DNA, RNA, and sometimes proteins, without introducing changes into their primary sequence, and with essential implications in organismal function and disease. Hence, translational epigenetics pays attention to the effect of chemical modifications on DNA [1], histones, non-histone proteins [2], and RNA [3], which in turn results in structural and functional changes of target molecules especially when these molecules may be of therapeutic interest. The objective of this review is to explore epigenetic modifications particularly relevant to RNA biology with a particular focus on those crucial in cardiovascular physiology and pathophysiology. For a general description of the regulation and effects of epigenetic changes occurring at the RNA level, the reader is redirected to recent articles that are more detailed in mechanistic terms [4,5,6,7].

Among the different RNA species, there are non-coding RNAs (ncRNAs), which are ribonucleic acid sequences that do not codify for proteins. They recently became of interest for their important regulatory function and perspective diagnostic-therapeutic potential. Conventionally, ncRNAs are classified by their length: 200 nucleotides are the cut off between long (lncRNAs) and short non-coding RNAs (sncRNAs). The latter group includes microRNAs (miRNAs) [8]. However, this classification is not predictive of the function of ncRNAs. Indeed, Amaral et al. proposed a new way to classify ncRNAs based on their biological roles [9]. Many non-coding RNAs, especially lncRNAs, can act as sponges for miRNAs [10], enhancer-associated factors [11], transcriptional repressors [12], and regulators of nuclear structures such as paraspeckles [13]. In this review, for simplicity, we will refer to the classification of ncRNA species according to their length.

Although several reviews have been written about the regulatory role of ncRNAs [14,15,16,17], our knowledge is still limited about epigenetic modifications occurring in ncRNAs in physiological and pathological conditions. It is well known, however, that RNA sequences can be the target of methyltransferases such as N6-adenosine methyltransferase-like 3 (METTL3). The most common modifications in RNA molecules are the methylation of adenine in position 1 (N^1^-methyladenosine, m^1^A) [18] and 6 (N^6^-methyladenosine, m^6^A) [19]. In particular, when m^6^A occurs at the 5′-AGG(m6)AC-3′ consensus sequence of some mRNAs [20], their stability is modulated [21] and their translation efficiency may be altered [22]. Interestingly, similarly to what occurs to 5-deoxymethylcytosine, the m^6^A of RNA can be oxidatively demethylated into N^6^-hydroximethyladenosine (hm^6^A) and N^6^-formyladenosine (f6A) which may modulate RNA–protein interaction affecting gene regulation [23]. These processes, catalyzed at the RNA level by the fat mass and obesity-associated protein (FTO) in the presence of iron oxide and α-ketoglutarate, are the expression of the profound interplay occurring between DNA, RNA, proteins, and cellular metabolism during the process of methylation and demethylation [23] (see Figure 1). Moreover, ribocytosines can be methylated at position 5 (5-methylcytosine, m5C) [24] by RNA methyltransferases such as the NOP2/Sun domain family (1–7) [25] but also by some DNA methyltransferase such as the DNA methyltransferase type 2 (DNMT2) [26] that initially was considered a DNA-specific methyltransferase. Lately, ten-eleven translocation proteins (TET, which act similarly on DNA and RNA molecules) were found converting RNA 5mC into 5-hydroxymethylcytosine (5hmC) which facilitates the translation of RNA molecules [27]. TET proteins are Fe(II) and α-ketoglutarate-dependent dioxygenases [28], further emphasizing the interconnection between cell metabolism and the epigenetic machinery controlling nucleic acid modifications (see Figure 2). Of note, methyl groups can be added on riboguanosine too, in particular at position 7′ generating 7-methylguanosine (m^7^G) [29]. This modification mainly occurs on capped [30] and recapped mRNAs and is mediated by canonical mRNA capping methyltransferase (RNMT), which regulates mRNA translation into proteins [31] (see Figure 3). For more detailed mechanistic insights, the readers will be directed to recent comprehensive reviews in which the molecular mechanism and biological functions are well described [32,33,34,35].

## 2. Methylation of Coding RNAs in Cardiovascular Diseases

Cardiovascular diseases (CVDs) are a pandemic problem that in 2017 caused around 17.7 million deaths worldwide [36]. These disorders are often triggered by chronic metabolic alterations such as those associated with insulin resistance, obesity, and diabetes, and are characterized by the presence of small and large vessels disease, heart failure, myocardial infarction, and stroke with or without ischemia, hypertension, coronary artery disease, valve disease, arrhythmias, cardiomyopathies (sporadic and congenital), and pericardial diseases [36].

Recently, the American Heart Association identified chronic heart failure as the most critical damaging condition for the heart in the aging population [37]. Myocardial infarction and pressure overload are the leading causes of heart failure as they may lead to cardiomyocytes hypertrophy and reduced myocardial pump function [38]. In this setting, the m^6^A RNA methyltransferase METTL3 seems to play a crucial role in eccentric cardiomyocyte remodeling [39]. One METTL3 target is, in fact, the mRNA encoding for the mitogen-activated protein kinases (MAPKs), resulting in the upregulation of the corresponding protein, which in turn induces gene expression, protein synthesis, and increase of cardiomyocyte size [39]. The work reported a higher level of m^6^A mRNA in mouse neonatal but not adult cardiac cells [39]. To explore further the role of m^6^A in this context, METTL3 knockout animals were exposed to pressure overload. Gene-deleted mice were resistant to cardiac hypertrophy indicating that the control of RNA methylation is important to prevent the development of heart failure likely by regulating RNA processing [39] (see Figure 4).

Recent work reported the correlation between heart failure, cardiac hypertrophy, and the position of RNA adenosine methylation in the transcript body. Interestingly, the physical enrichment of m^6^A RNA methylation in specific transcript regions was correlated to the efficiency of protein translation, possibly due to a differential effect on the interaction between mRNA and ribosomes [40]. Specifically, when the m^6^A modification was present in the 5′UTR and coding region of mRNAs, the translation of regulatory components involved in mitochondrial function and cellular metabolism was preferred. On the contrary, whether the methylation occurred at 3′UTR, acetyl-Co, glycerol biosynthesis, and the positive regulation of protein dephosphorylation was advantaged [40].

Another study showed that the FTO protein decreased in the failing heart resulting, in the presence of hypoxia, in RNA hypermethylation of adenosine, increasing the content of m^6^A [41]. In this study, the downregulation of the FTO protein correlated with alterations in calcium dynamics and, as a consequence, the modification of cardiomyocytes contraction, exacerbating in arrhythmic events that are frequently observed in heart failure [41]. The role of m^6^A in mRNA has been studied in dilated cardiomyopathy and failing heart. This study revealed an increased METTL3 activity and the preponderant presence of RNA transcripts enriched in m^6^A compared to healthy myocardium. The consequence of this enrichment is higher mRNA instability and reduced expression of genes involved in hypertrophic cell growth. This evidence highlighted an unprecedented correlation between the abundance of m^6^A in RNA and the increase in cellular volume [42]. In failing hearts, one example is given by the m^6^A hypermethylation of mRNA encoding for myosin regulatory light chain 2 (Myl2), resulting in lower protein levels than healthy controls [42], suggesting in this case for a destabilizing effect of m^6^A (see Figure 5). The destabilizing effect, however, seems context-dependent or, perhaps, associated with specific pathophysiological conditions. Often, the presence of atherosclerotic plaques correlates with inflammation and macrophages infiltration [43]. Recent work reported an epigenetic mechanism regulating macrophage function [44]. In this context, METTL3 methylated the signal transducer and activator of transcription 1 (STAT1) mRNA in M1 macrophages [44]. The regions of STAT1 more prone to modification were the coding sequence and the 3′-untranslated region, overall increasing the relative mRNA stability [44]. As a result, the methylation of STAT1 mRNA, increasing STAT1 translation and activity, seemed to drive M1 macrophage polarization, defining a potentially novel anti-inflammatory signaling pathway [44] (see Figure 4).

A recent observation suggested a role for m^6^A RNA during autophagy in cardiomyocytes [45] subjected to hypoxia and reoxygenation. The latter is a condition that upregulates METTL3, resulting in methylation of m^6^A in the mRNA encoding for the transcription factor EB (TEFB), which is involved in lysosomal biogenesis [45], and repression of protein synthesis [45]. This process eventually leads to reduced autophagy and increased apoptosis after hypoxia damage [45]. Interestingly, an RNA demethylase named ALKB homolog 5 (ALKBH5) removes the methyl group in TEFB mRNA adenosine. ALKBH5, in physiological conditions, is transcriptionally activated by TEFB itself, which binds the ALKBH5 promoter, realizing a positive feedback loop controlled at the RNA methylation level [45] (see Figure 5).

Regarding cytosine methylation, in endothelial cells, NOP2/Sun domain family member 2 (Nsun2) is known to methylate the transcript encoding the intercellular adhesion molecule 1 (ICAM-1), increasing the corresponding protein’s synthesis and enhancing leukocytes adhesion to the endothelial layer [46]. Noteworthy, in vascular smooth muscle cells and endothelium, the downregulation of Nsun2 protects against the exacerbation of inflammation, suggesting a pivotal role for mRNA methylation in the pathogenesis of atherosclerosis [46,47]. Whether a differential methylation process is active in endothelial or vascular smooth muscle cells is at present unknown. However, dysmetabolic conditions or metabolic risk factors are well documented as able to modify the epigenomic landscape [48]. Hence, we may postulate that metabolic alterations associated with atherosclerosis may be implicated in the regulation of RNA processing leading to the onset of specific pathophysiological conditions.

Hyperhomocysteinemia (HHcy) is a metabolic alteration often associated with inflammation and frequently detected in cardiovascular and chronic diseases. Of interest, the effect of HHcy seems mediated by Nsun2, which, in T cells, methylates the mRNA encoding for interleukin (IL)-17A [49]. The methylation occurs on m5C present in the mRNA coding region and results in enhanced IL-17A translation suggesting that the modification is pivotal to protein synthesis, indirectly contributing to the onset of inflammation [49]. Indeed, IL-17A has been correlated with atherosclerotic plaque formation [50]. Consistently, high levels of IL-17A have been found in B and T cells, in macrophages, and plasma cells present within atherosclerotic lesions. Moreover, IL-17A synthesis has been positively associated with plaque vulnerability [50] (Figure 6).

## 3. Non-Coding RNA and Their Promoter Methylation in Cardiovascular Diseases

Inflammatory signals are at the basis of all chronic diseases, and they are often triggered by dysmetabolic conditions associated with aging, a process widely defined as inflammaging [51,52,53,54]. In an experimental model of prolonged hyperglycemia and cardiomyopathy, high levels of the metastasis-associated lung adenocarcinoma transcript-1 lncRNA (MALAT1) were reported [55]. In this condition, the inhibition of phosphodiesterase 5, through the administration of sildenafil, resulted in an increase of nitric oxide that normalized MALAT1 levels in cardiomyocytes by an unknown mechanism [55]. Interestingly, the MALAT1 transcript has multiple sites of adenosine methylation, especially on consensus sequences such as GGACU, AGACA, and GAACC [56]. Such modification introduced conformational changes in the RNA molecule, promoting its association with multiple RNA binding proteins [57]. As an example, the association with heterogeneous nuclear ribonucleoprotein G [58] induced pre-mRNA processing and alternative splicing [59].

Moreover, myeloid cells from MALAT1-deficient mice displayed a higher adhesiveness to atherosclerotic lesions and lower levels of MALAT1 expression in human plaques could be related to worse prognosis caused by the infiltration of inflammatory CD45 + cells [60]. In addition, a role for MALAT1 has been proposed in myocardial infarction (MI) where its expression is often upregulated [61]. In fact, the knockdown of MALAT1, promoted the progression of cardiomyocytes through the cell cycle and suppressed apoptosis via modulation of the miR-200a-3p/PDCD4 axis [61] (see Figure 4).

In a series of in vitro experiments performed by using human umbilical vein endothelial cells in the presence of oxidized low-density lipoproteins and platelet-derived growth factor, miR-125b was downregulated whereas podocalyxin (PODXL), a member of the cluster of differentiation 34 of sialomucins, resulted as upregulated [62]. Interestingly, miR-125b is repressed in atherosclerosis and, consequently, to increased PODXL activity, vascular endothelium expressed pro-adhesive proteins such as cadherin and ICAM-1 [62]. On the contrary, in vascular smooth muscle cells (VSMCs), it has been demonstrated that the upregulation of miR-125b correlates with atherosclerosis, possibly through the activation of transcription factor SP7, which regulates the transdifferentiation of VSMCs into osteoblast-like cells. The methylation of pre-miR-125b seems to be an essential mechanism of regulation in this context. Methylation occurring by NOP2/Sun domain family member 2 (Nsun2) represses the formation of mature miR-125b by adding a methyl group on the adenosine present in RRACH and AAC motifs [63].

In the context of regulatory processes, it emerged clearly that positive and negative feedback loops are present during the process of RNA maturation in which methylation might play a role at different levels. For example, the expression of miRNAs is frequently regulated by methylation of the corresponding DNA locus. The transcription of miR-200 family members requires, in fact, DNA demethylation at the promoter level and is enhanced by reducing the activity of DNMTs [64]. In a rat model of cardiac fibrosis and abdominal aortic constriction [64], miR-200b inhibited the expression of LC3BII/I, which is a trigger signal in cardiac myofibroblast conversion, a process that increases autophagy.

HHcy is a known risk factor for ischemic stroke [65], in which methylation occurs on non-coding RNA promoters [66] and specific mRNAs, as previously described [49]. It has been shown that in the presence of HHcy, the lncRNA H19 promoter results as methylated and plays an essential regulatory role [66]. In fact, in rodents, the differentially methylated domain located in H19 promoter results as hypomethylated in the presence of HHcy, altering the expression of H19 and that of insulin growth factor 2 (IGF2) which is located nearby [66]. In rats, HHcy determines tissue-specific alterations of methylation at the level of the H19 promoter. In particular, H19 promoter methylation was observed in rat aorta, while lower levels were found in the liver [66]. Interestingly, the expression of H19 inhibits S-adenosylhomocysteine hydrolase (SAHH), which hydrolyzes S-adenosylhomocysteine (SAH) that, in turn, acts as an inhibitor of S-adenosylmethionine (SAM)-dependent methyltransferases [67]. When SAH binds the DNA methyltransferase type 3 (DNMT3B), which is involved in de novo DNA methylation [68], this enzyme loses its capacity to add the methyl group on its targets [67]. One of these targets is the coding and promoter region of the non-coding transcript 1 (NCTC1) that in turn, controls H19 expression determining the so-called promoter competition [67]. As a result, H19 expression changes altering expression of miR-29b [69] and miR-877-3P [70] with consequences for endothelial function [71] and cardiomyocyte apoptosis [69] leading to a higher risk of abdominal aortic aneurism [72] and adverse outcome in myocardial ischemia-reperfusion injury [73].

In summary, the methylation of H19 locus has effects on H19 expression and function and interferes with the activity of DNA methylases determining changes in the expression level of other genes with potential consequences on cardiovascular homeostasis (see Figure 7).

Other ncRNA species are regulated by methylation at the genomic level with consequences on cardiovascular homeostasis. In vivo, the miR-145 expression has an impact on atherosclerosis and pulmonary hypertension. High levels of this miR protect blood vessels from plaque formation and hypertension [74,75]. Hence, the methylation of miR-145 promoter downregulated the expression of the cis-regulated miR preventing its adverse action on the nuclear factor of activated T cells 1 (NFATc1) and CD137 [76]. As a result, VSMCs expressed more nucleotide-binding oligomerization domain-like receptor protein 3 (NLRP3) that, in turn, activated interleukin 1β (IL-1β) secretion promoting intravascular inflammation [76].

Additionally, miR-145 regulates ubiquitin-like containing PHD and RING finger domains 1 (UHRF1), which interact with DNA methyltransferase 1 (DNMT1), suggesting an indirect link between miRNA modulation and DNA methylation [77]. DNMT1 was found to methylate miR-145 promoter realizing a negative feedback loop [76] (Figure 8).

## 4. Future Perspective of Epigenetics as a Treatment and Diagnostic Marker

Several studies have demonstrated the role of methylation in the epigenetic regulation of lncRNAs expression during cardiac development and repair. Here we provide a few recent examples of this very recent aspect discussing whether they could be developed as therapeutic targets.

An example is given by the lncRNA named cardiomyocyte proliferation regulator (CPR) that, if downregulated, induces cell cycle activation and decreases scar formation after cardiac injury [78]. CPR regulates the expression of minichromosomal maintenance 3 (MCM3) gene by recruiting DNMT3A and CpG island methylation on the MCM3 promoter [78]. This finding suggests that lncRNA might be the target of a new treatment against myocardial infarction. Another lncRNA, called antisense non-coding RNA in the cyclin-dependent kinase inhibitors (INK4) locus (ANRIL), revealed a potential pathophysiological role in overweight newborns (24.6% fat mass at birth) [79]. In this study, umbilical cords were collected at birth, and nine CpG sites were investigated. After nine years, blood pressure, heart rate, and pulse wave velocity of the donors were measured [79]. It was found that ANRIL promoter methylation correlated with heart rate and pulse wave velocity, while blood pressure seemed not linked to ANRIL [79]. This study suggests that non-coding RNAs may be an indicator of clinical risk for CVD development in childhood. 

## 5. Conclusions

Although the field of epigenetics is rushing forward, most of the epigenetic studies on CVDs and/or chronic diseases are still focused on DNA methylation [80], histone code modifications [81], and the regulation of protein expression and function by a plethora of ncRNAs [82] rather than on the quality, quantity, and role of RNA epigenetic modifications.

Recent literature highlighted the controversial role of RNA methylation, occurring mainly on adenosines, and resulting in increased mRNA instability, repressed protein synthesis [42,45], or its stabilizing regulatory loops between mRNAs and lncRNAs [39,42,44,61]. On the other hand, the methylation of mRNA cytosines by Nsun2 has a stabilizing effect determining an increase in transduction [49]. It must be said, however, that most of the studies about RNA molecules modified as m^6^A and m^5^C come from cancer biology leaving cardiovascular diseases relatively unexplored [83,84,85].

At present, several lncRNAs and miRNAs have been investigated in differential pathophysiological contexts including hypoxia [61], cardiac regeneration [86], atherosclerosis [74], and in the presence of altered blood pressure [75], but little is known about the consequence of epigenetic modifications on their structure, regulation, and function (see Table 1). The physiological and pathophysiological consequences of RNA modification in adenosine and cytosine and their involvement in CVDs remains mostly unclear [39,41,78]. However, this new investigational direction promises to provide insights into unique mechanisms leading to potential new therapeutic discoveries.

## Figures and Tables

**Figure 1 ijms-21-00509-f001:**
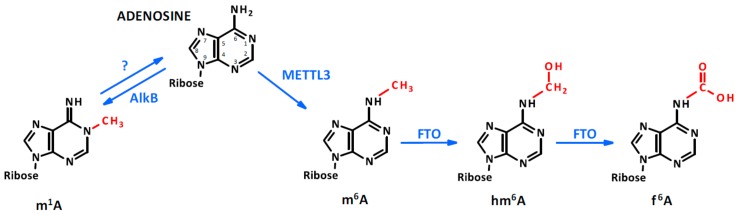
Adenosine methylation. Adenosine can be methylated by the N6-adenosine methyltransferase-like 3 (METTL3) in the sixth position. In the presence of Fe(II) and α ketoglutarate, the dioxygenase fat mass and obesity-associated (FTO) protein oxidates the methyl group, generating hm6A and f6A. Meanwhile, the methylation of adenosine in the first position occurs through an unclear process which can be reverted by AlkB demethylase.

**Figure 2 ijms-21-00509-f002:**
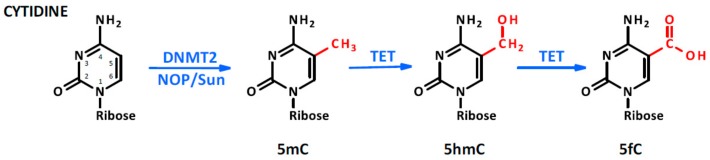
Cytosine methylation. The methylation occurs in the fifth position of the molecule as a consequence of DNA methyltransferase type 2 (DNMT2) or NOP/Sun family member activity. Moreover, in the presence of Fe(II) and α ketoglutarate, TET protein oxidates the methyl group into hydroxymethyl and formyl groups.

**Figure 3 ijms-21-00509-f003:**
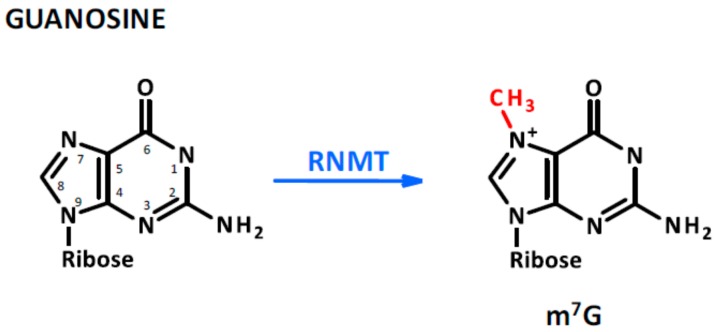
Guanosine methylation. This modification is not as abundant as that of adenosine and cytosine; however, it is found on capped and re-capped mRNAs as a consequence of mRNA capping methyltransferase (RNMT) protein activity.

**Figure 4 ijms-21-00509-f004:**
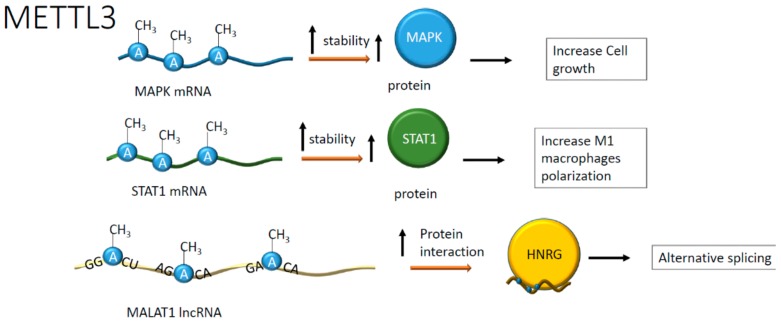
Positive effects on RNA stability and protein interaction upon adenosine methylation. Positive effects have been reported on mitogen-activated protein kinases (MAPKs) and signal transducer and activator of transcription 1 (STAT1) mRNAs whose translation was enhanced. In addition, the methylation in specific ApC sequences on metastasis-associated lung adenocarcinoma transcript-1 (MALAT1) increases the interaction with heterogeneous nuclear ribonucleoprotein G (HNRG) proteins contributing to alternative splicing.

**Figure 5 ijms-21-00509-f005:**
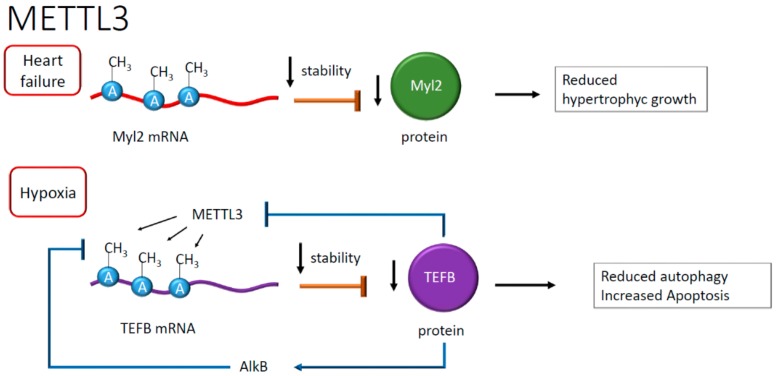
Negative effects on RNA stability after adenosine methylation. Reduced mRNA stability has been reported in heart failure (HF) and the presence of hypoxia. In HF, the downregulation of myosin regulatory light chain 2 (Myl2) reduces hypertrophic growth, whereas, in hypoxia, lower levels of transcription factor EB (TEFB) have negative feedback on its mRNA methylation, reflecting the reduction of autophagy and an increase of apoptosis.

**Figure 6 ijms-21-00509-f006:**
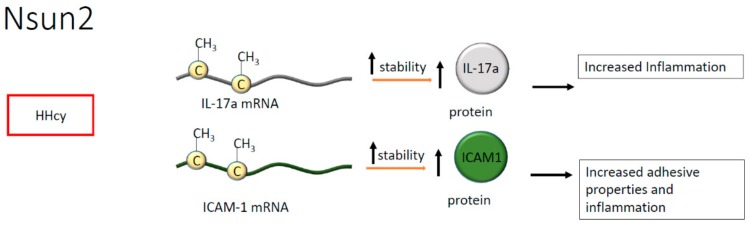
Methylation of cytosine by NOP2/Sun domain family member 2 (Nsun2) in hyperhomocysteinemia (HHcy) increases the stability of interleukin (IL)-17a and intercellular adhesion molecule 1 (ICAM1) mRNAs, indicating a global increase of inflammation.

**Figure 7 ijms-21-00509-f007:**
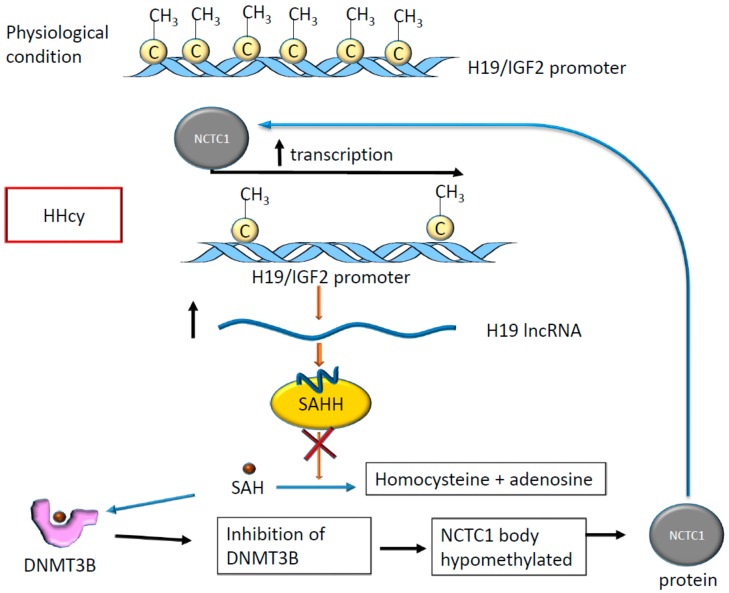
The hypomethylation of the H19 promoter in HHcy alters the activity on the S-adenosylhomocysteine hydrolase (SAHH) enzyme, which is inhibited by the binding with H19. The accumulation of S-adenosylhomocysteine (SAH) inside cells increases binding between SAH and DNA methyltransferase type 3 (DNMT3B), determining DNMT3B inactivation. As a result, the NCTC1 gene is not methylated, acting on the H19 promoter to determine a positive transcriptional feedback loop.

**Figure 8 ijms-21-00509-f008:**
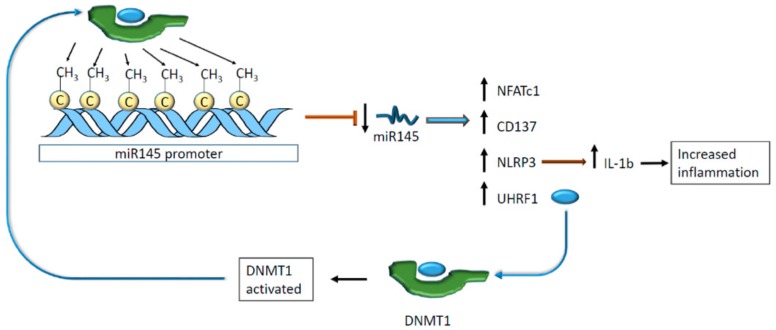
miR145 promoter hypermethylation decreases miR145 levels. This phenomenon results in an upregulation of different genes, enhancing the transcription of cytokines related to inflammation. Moreover, low levels of miR145 allow the transduction of UTRF1, which is an activator of DNA methyltransferase 1 (DNMT1) that, in turn, methylates miR145 promoter activating a negative feedback loop.

**Table 1 ijms-21-00509-t001:** RNAs modification in cardiovascular diseases (CVDs) and their effects. The table summarizes the most relevant epigenetic modifications occurring on RNA molecules or in their cognate promoter regions. Data are assembled according to distinct cardiovascular disease or pathophysiological conditions.

CVDs	Nucleic Acid Involved	Enzyme	Base	Effects	Reference
Cardiac hypertrophy	mRNA of MAPK	METTL3	A	Increased size of cardiomyocytes	[39]
Dilated cardiomyopathy	Global RNA	METTL3	A	Cell growth arrest	[42]
Cardiomyopathy after hypoxia/reperfusion injury	LncRNA (MALAT1)	METTL3	A	Apoptosis	[61]
Cardiac hypoxia	mRNA of TEFB	METTL3	A	Increased apoptosis and reduced autophagy	[45]
Ischemic stroke in the presence of HHcy	DNA promoter of H19/IGF2	DNMT3B	C	Alteration of H19 and IGF2 expression; induction of apoptosis in cardiomyocyte	[66,69]
Atherosclerotic plaque formation	mRNA	METTL3	A	Induction of M1 macrophage polarization; increased inflammation	[44]
	LncRNA (MALAT1)	METTL3	A	Increased vascular adhesiveness	[60]
	MiRNA (microRNA125b)	Nsun2	C	Increased cell adhesion on VCAM	[63]
	DNA promoter of microRNA145	DNMT3B	C	Increased cell adhesion and inflammation	[74,75]
Atherosclerotic plaque formation in association with HHcy	mRNA of IL-17A	Nsun2	C	Increased expression of pro-inflammatory cytokines	[49,50]

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
