# Peer review of "Epigenetic Signaling and RNA Regulation in Cardiovascular Diseases"

_ijms, 2020, doi:10.3390/ijms21020509_

Round 1

Reviewer 1 Report

In their review article  ‘Epigenetic Signalling and RNA Regulation in Cardiovascular Diseases,’ Mongelli et al. are summarizing the current knowledge on epigenetic RNA regulation with a focus on non-coding RNA in cardiovascular disease. The authors furthermore state to provide information on the translational potential of this novel field of DNA-RNA-protein interplay.

Overall, the article approaches an important and timely subject with promising initial results being reported on the one hand and a vast variety of unanswered questions attracting scientist to explore this field. In this respect, the article’s theme is likely to attract scientists’ attention, especially seeing its translational potential.

However, while the authors introduce in the abstract, that the article will cover translational aspects and potential clinical applicability, enthusiasm for the manuscript as it currently is, is diminished by the fact that these translational aspects are only mentioned to a minute extend. Furthermore, the manuscript is somewhat cumbersome to follow and would benefit from a substantial increase in structure such as more headings and sub-headings and fitting single mechanisms or diseases into a defined paragraphs. Similarly, some sentences are very long and hard to follow. Also in this case more structure and shorter sentences would improve the reading experience.

1) In lines 73 and 74 the authors write: ‘For more detailed mechanistic insights, the readers will be directed to recent comprehensive reviews’. The next paragraph covers implementations of the mechanistic insights in cardiovascular disease. It might be beneficial to provide references for and possibly brief comments on the ‘recent comprehensive reviews’ already at this stage.

2) In line 101 the authors refer to ‘another work’. No reference is given, though. Most likely the authors are referring to a citation from within reference 35. I suggest a paraphrasing of the paragraph dealing with citation 35 to clarify.

3) The paragraph ‘mRNAs and ncRNAs methylation in cardiovascular disease’ is somewhat difficult to follow. It would benefit from more structure; i.e. the covered aspects related to cardiovascular disease could be structured by disease entity such as 1. Heart failure, 2. Coronary artery disease etc.

4) Following up on point 3) above, the paragraph starting in line 139 needs a header or be defined by a subject that it is covered under. Given that MALAT1 is associated with atherosclerosis, it could be covered under ‘atherosclerosis’ or ‘coronary artery disease’. Alternatively, could there be a structure covering mRNAs, lncRNAs, miRNAs (miRNAs are mentioned from line 155 onwads – again without a being fit into a structure)?

5) In line 139 the authors state that ‘inflammatory signals are at the basis of all chronic diseases’. Here, I suggest to either provide references for this strong statement or soften the ‘all’ down. Furthermore, this introductory sentence lets the reader expect the paragraph to cover inflammatory disease, which is not as much the case as one would expect.

6) In line 216 a paragraph starts, seems to refer to previously mentioned SAH and DNMT3B. These have not been introduced; again better structure would be helpful.

7) Lines 222 to 224 are a good example of how the end of a subject or paragraph can be summarised. It would be beneficial for the manuscript if such summaries were included more often.

8) The paragraph from line 230 to line 246 is well written. Again, it would be beneficial to structure it into a sub-heading to determine its place within the whole manuscript.

9) The paragraph from line 247 to 261 need a header – maybe with respect to ‘therapeutics’?

Author Response

The authors would like to thank the Reviewer for the helpful suggestions.

As suggested we added the references, where the insights can be found, line 74-75. We added the reference which was the same previously cited,line 103 The literature on ncRNAs methylation and DNA methylation of ncRNAs promoter in CVDs is about preliminary and too scattered to create specific paragraphs on single disease entities. During the writing the authors noticed the controversial effects of METTL3 on RNAs stability. In fact, in some cases the enzyme increases the stability but in other deceases. Until now, none has reported this obsevation. in addition, we created a table (line 292) in order to list and compare what is known for single CVDs in this field. The reference has been already cited, however, we cited new article regarding the inflammation in chronic diseases (line 142). The text has been modified in order to send the reader to other reviews about the specific argument (line 142). We modified the text and introduced briefly the argument, line 223. We modified some parts of the article (line 221, 228,170, 223, 211) . We made the conclusion after at the end of paragraphs, line 170. We added it, line 254.

Reviewer 2 Report

Mongelli et al reviewed RNA epigenetic regulation in cardiovascular disease. The manuscript was well and succinctly written, and the figures were nicely illustrated. My only comment is to better structuralize the manuscript. It is hard to tell where the introduction section ends and I only saw two sections “mRNAs and ncRNAs methylation in cardiovascular diseases” and “Methylation at ncRNA promoters and its consequences for CVDs” in the main text even though those two titles seem partially overlapped. Please further sectionalize the main text, e.g., separate mRNA and ncRNA sections.

Author Response

The authors thank the reviewer for the valuable suggestions. While the article was submitted the authors noticed the newly reported article about the controversial effects of METTL3 on RNAs stability. In this revised form we reorganized text to describe more in detail  the activity of METTL3 and Nsun2 regardless specific RNA types. In addition, a table has been added summarizing  the modification occurring on RNA or at DNA promoter level  in apecific CVDs.

Round 2

Reviewer 1 Report

Neither did the authors privide a point-by-point response to the questions and issues nor was a document provided highlighting the changes as it is common practice.This makes it very hard to provide a high-quality review of the revised manuscript.

It cannot be expected from a reviewer that from a new manuscript version without any markups all changes are tracked on one's own, in particular after a major revision with focus on a new structure of the manuscript was suggested in the first review process.

With respect to my evaluation of a major revision, the authors' reply is insufficient.

I am happy to provide a new review once a structured revision has been provided.

I hope for the authors' understanding.

Author Response

Answers to reviewers questions:
Reviewer 1
The authors thank the Reviewer for His/Her helpful suggestions. However, we would like to point out that it is impractical to catalogue this very recent and cogent matter according to pathology as suggested. In our opinion, in fact, there is not sufficient body yet to create disease-specific subchapters. Nevertheless, in the revised manuscript version a new table (table 1) has been
created summarising the state of the art about specific pathophysiological conditions and epigenetic modifications of selected non-coding RNAs. In addition, all other Reviewers’ observations have been addressed as reported in the following point-to-point letter.
1. As suggested new references have been added, see line 74-75.
2. During revision, the authors considered of interest the controversial effects of METTL3 on RNAs stability. A specific paragraph now addresses this point. In addition, as stated above, a new table has been added listing non-coding RNA modification associated, when possible, with specific cardiovascular diseases (line 291).
3. A new article has been cited about the role of inflammation determining epigenetic modifications in the presence of chronic disease (line 142).
4. Text has been modified indicating more clearly to readers about recent review articles addressing specifically addressing the matter (line 142).
5. Text has been rewritten briefly discussing the suggested matter (line 223).
6. Lines 170, 211, 221, 223, 228 have been partly rewritten according to suggestion.
7. A conclusion paragraph has been introduced at the end of the article (line 270).
8. We added it, line 254.

Round 3

Reviewer 1 Report

In their review article  ‘Epigenetic Signalling and RNA Regulation in Cardiovascular Diseases,’ Mongelli et al. are summarizing the current knowledge on epigenetic RNA regulation with a focus on non-coding RNA in cardiovascular disease. The authors furthermore state to provide information on the translational potential of this novel field of DNA-RNA-protein interplay.

The authors have replied (blue) to my previously raised points (black) as follows and I am replying below (red) to the revised version of the manuscript. Throughout the point-by-point answers, wrong line labelling is used and the answers from point 3 on have the wrong numbering, both making it hard to track the changes and match the authors’ answers.

Overall, the article approaches an important and timely subject with promising initial results being reported on the one hand and a vast variety of unanswered questions attracting scientist to explore this field. In this respect, the article’s theme is likely to attract scientists’ attention, especially seeing its translational potential.

However, while the authors introduce in the abstract, that the article will cover translational aspects and potential clinical applicability, enthusiasm for the manuscript as it currently is, is diminished by the fact that these translational aspects are only mentioned to a minute extend. Furthermore, the manuscript is somewhat cumbersome to follow and would benefit from a substantial increase in structure such as more headings and sub-headings and fitting single mechanisms or diseases into a defined paragraphs. Similarly, some sentences are very long and hard to follow. Also in this case more structure and shorter sentences would improve the reading experience.

The authors thank the Reviewer for His/Her helpful suggestions. However, we would like to point out that it is impractical to catalogue this very recent and cogent matter according to pathology as suggested. In our opinion, in fact, there is not sufficient body yet to create disease-specific subchapters. Nevertheless, in the revised manuscript version a new table (table 1) has been created summarising the state of the art about specific pathophysiological conditions and epigenetic modifications of selected non-coding RNAs. In addition, all other Reviewers’ observations have been addressed as reported in the following point-to-point letter.

My suggestion was to provide more structure to the manuscript to enable the reader to follow the content. As one possible example I suggested to structure it by pathology. The authors argue that this is impractical and that is probably correct. In this case, it is the authors’ responsibility to work on alternative ways to provide more structure, rather than simply provide a table and argue against my example. Since my major point of criticism has not been met, which is to provide substantially more structure to the manuscript – in which ever way – I can only repeat myself with a recommendation to major revision in this respect and provide the following overall judgement:

Overall, the article approaches an important and timely subject with promising initial results being reported on the one hand and a vast variety of unanswered questions attracting scientist to explore this field. In this respect, the article’s theme is likely to attract scientists’ attention, especially seeing its translational potential.

However, while the authors introduce in the abstract, that the article will cover translational aspects and potential clinical applicability, enthusiasm for the manuscript as it still is after the revision, is diminished by the fact that these translational aspects are still only mentioned to a minute extend. Furthermore, the manuscript is still somewhat cumbersome to follow, since no significant structural changes have been made.

1) In lines 73 and 74 the authors write: ‘For more detailed mechanistic insights, the readers will be directed to recent comprehensive reviews’. The next paragraph covers implementations of the mechanistic insights in cardiovascular disease. It might be beneficial to provide references for and possibly brief comments on the ‘recent comprehensive reviews’ already at this stage.

As suggested new references have been added, see line 74-75.

This improves the manuscript

2) In line 101 the authors refer to ‘another work’. No reference is given, though. Most likely the authors are referring to a citation from within reference 35. I suggest a paraphrasing of the paragraph dealing with citation 35 to clarify.

During revision, the authors considered of interest the controversial effects of METTL3 on RNAs stability. A specific paragraph now addresses this point. In addition, as stated above, a new table has been added listing non-coding RNA modification associated, when possible, with specific cardiovascular diseases (line 291).

This improves the manuscript, but the new table should have a legend.

3) The paragraph ‘mRNAs and ncRNAs methylation in cardiovascular disease’ is somewhat difficult to follow. It would benefit from more structure; i.e. the covered aspects related to cardiovascular disease could be structured by disease entity such as 1. Heart failure, 2. Coronary artery disease etc.

A new article has been cited about the role of inflammation determining epigenetic modifications in the presence of chronic disease (line 142).

I am not able to fit this answer with any of my raised issues. Please clarify. Again, the point-by-point answer seems not to show the appropriate thoroughness suitable for a peer-reviewed journal.

4) Following up on point 3) above, the paragraph starting in line 139 needs a header or be defined by a subject that it is covered under. Given that MALAT1 is associated with atherosclerosis, it could be covered under ‘atherosclerosis’ or ‘coronary artery disease’. Alternatively, could there be a structure covering mRNAs, lncRNAs, miRNAs (miRNAs are mentioned from line 155 onwads – again without a being fit into a structure)?

Text has been modified indicating more clearly to readers about recent review articles addressing specifically addressing the matter (line 142).

This answer – titled with “4” by the authors most likely is supposed to be an answer to point 5 instead. In this case the point is met

5) In line 139 the authors state that ‘inflammatory signals are at the basis of all chronic diseases’. Here, I suggest to either provide references for this strong statement or soften the ‘all’ down. Furthermore, this introductory sentence lets the reader expect the paragraph to cover inflammatory disease, which is not as much the case as one would expect.

Text has been rewritten briefly discussing the suggested matter (line 223).

This answer – titled with “5” by the authors most likely is supposed to be an answer to point 6 instead. In this case the point is met.

6) In line 216 a paragraph starts, seems to refer to previously mentioned SAH and DNMT3B. These have not been introduced; again better structure would be helpful.

Lines 170, 211, 221, 223, 228 have been partly rewritten according to suggestion.

This answer – titled with “6” by the authors most likely is supposed to be an answer to point 7 instead. ...

7) Lines 222 to 224 are a good example of how the end of a subject or paragraph can be summarised. It would be beneficial for the manuscript if such summaries were included more often.

A conclusion paragraph has been introduced at the end of the article (line 270).

This answer – titled with “7” by the authors most likely is supposed to be an answer to point 8 instead. In this case, I am not able to see any changes as indicated by the authors. Please clarify.

8) The paragraph from line 230 to line 246 is well written. Again, it would be beneficial to structure it into a sub-heading to determine its place within the whole manuscript.

We added it, line 254.

This answer – titled with “8” by the authors most likely is supposed to be an answer to point 9 instead. There is a new header in line 251. I suppose that is one the authors are referring to?

9) The paragraph from line 247 to 261 need a header – maybe with respect to ‘therapeutics’?

Author Response

To the Reviewer,

we would thank for your helpful suggestions improving quality of our article.  As you suggested, the whole text has been reorganized as follow:

An updated introduction: providing insights about the most common epigenetic modifications occurring to RNAs. A new chapter about the effect of RNA methylation on selected mRNAs associated with cardiovascular diseases (CVDs). Description of the effect of methylation on non-coding RNA transcripts associated with CVDs or their genomic DNA promoter regions. A comment about RNA epigenetic modifications as CVDs biomarkers or future therapeutic targets An updated conclusion, highlighting the controversial role of m6A RNA on the stability of RNA molecules.

In addition,  a very recent article ( Berulava et al., December 2019; line 107-115)has been added reporting about the role of m6A in different mRNA regions and its impact on the exacerbation of cardiac hypertrophy.

Round 4

Reviewer 1 Report

Still no markups to find the changes compared to the last version - where are the changes in the introduction, as stated by the authors? Still no legend to the table Stil no point-by point answers to my previously raised issues

Author Response

To the Reviewer,

The authors would like to thank the Reviewer for His/Her suggestions strengthening our work. During this round of revision the following points have been addressed:

In new article version line 73-74 become 75-75 and we added new citations numbered 32,33,34 and 35 (highlighted in yellow) We added a short the table legend (yellow) and line 291 is now numbered 284. According to suggestion, in order to make it easier to read the whole text is now divided in two parts: coding and non-coding RNA. A new header has been added and line 142 is now numbered 184. Also, we inserted new references numbered 51, 52, 53. We provided a brief explanation in the paragraph starting at line 182 (yellow) The specific point has been clarified in line 228 (highlighted in yellow)

7 and 8. In the revised text, the light blue colour has been used to describe specific biological effects. Changes have been introduced in lines: 53-55; 100-102; 104-106; 118-120; 123-124; 134-136; 146-147; 162-165; 177-178; 192-193; 198-200; 219-220; 234-237; 251-253; 268; 275-276. Moreover, a new comment about recent article has been added in lines 107-115 (yellow)

the header refers to the paragraph “future perspective….” line 263 of the newly revised article (highlighted in yellow).